# Phylogenetic and Haplotype Network Analyses of *Diaporthe eres* Species in China Based on Sequences of Multiple Loci

**DOI:** 10.3390/biology10030179

**Published:** 2021-03-01

**Authors:** Chingchai Chaisiri, Xiangyu Liu, Yang Lin, Yanping Fu, Fuxing Zhu, Chaoxi Luo

**Affiliations:** 1Key Lab of Horticultural Plant Biology, Ministry of Education, Huazhong Agricultural University, Wuhan 430070, China; chaisiri.ch@gmail.com (C.C.); xiangyuliu@webmail.hzau.edu.cn (X.L.); 2Key Lab of Crop Disease Monitoring and Safety Control in Hubei Province, Huazhong Agricultural University, Wuhan 430070, China; yanglin@mail.hzau.edu.cn (Y.L.); yanpingfu@mail.hzau.edu.cn (Y.F.); 3College of Plant Science and Technology, Huazhong Agricultural University, Wuhan 430070, China; zhufuxing@mail.hzau.edu.cn

**Keywords:** GCPSR, *Phomopsis eres*, phylogeny, population connectivity, species delimitation

## Abstract

**Simple Summary:**

*Diaporthe eres* is one of the most serious plant pathogenic fungi that affect many economically important plants. It can cause rootstock death, stem canker, stem necrosis, dead branch, shoot blight, fruit rot, leaf spot, leaf necrosis, and umbel browning. In general, morphological and molecular characterization using multiple loci sequences were performed for the identification of *Diaporthe* species. However, there are morphological differences due to culture conditions, and the taxonomy of species of *Diaporthe* is unclear because the phylogeny based on different genes gives different tree topologies. In this study, we evaluate the phylogenetic relationships and population diversity among *D. eres* and other *Diaporthe* species. Our results showed that phylogenetic analyses from concatenated multi-locus DNA sequence data could resolve the *D. eres* species. Furthermore, haplotype network analysis showed that no correlation existed between population diversity and distribution or hosts across China. These results could improve our understanding of the epidemiology of *D. eres* and provide useful information for effective disease management.

**Abstract:**

*Diaporthe eres* is considered one of the most important causal agents of many plant diseases, with a broad host range worldwide. In this study, multiple sequences of ribosomal internal transcribed spacer region (ITS), translation elongation factor 1-α gene (*EF1-α*), beta-tubulin gene (*TUB2*), calmodulin gene (*CAL*), and histone-3 gene (*HIS*) were used for multi-locus phylogenetic analysis. For phylogenetic analysis, maximum likelihood (ML), maximum parsimony (MP), and Bayesian inferred (BI) approaches were performed to investigate relationships of *D. eres* with closely related species. The results strongly support that the *D. eres* species falls into a monophyletic lineage, with the characteristics of a species complex. Phylogenetic informativeness (PI) analysis showed that clear boundaries could be proposed by using *EF1-α*, whereas ITS showed an ineffective reconstruction and, thus, was unsuitable for speciating boundaries for *Diaporthe* species. A combined dataset of *EF1-α*, *CAL*, *TUB2*, and *HIS* showed strong resolution for *Diaporthe* species, providing insights for the *D. eres* complex. Accordingly, besides *D. biguttusis*, *D. camptothecicola*, *D. castaneae-mollissimae*, *D. cotoneastri*, *D. ellipicola*, *D. longicicola*, *D. mahothocarpus*, *D. momicola*, *D. nobilis*, and *Phomopsis fukushii*, which have already been previously considered the synonymous species of *D. eres*, another three species, *D. henanensis*, *D. lonicerae* and *D. rosicola*, were further revealed to be synonyms of *D. eres* in this study. In order to demonstrate the genetic diversity of *D. eres* species in China, 138 *D. eres* isolates were randomly selected from previous studies in 16 provinces. These isolates were obtained from different major plant species from 2006 to 2020. The genetic distance was estimated with phylogenetic analysis and haplotype networks, and it was revealed that two major haplotypes existed in the Chinese populations of *D. eres*. The haplotype networks were widely dispersed and not uniquely correlated to specific populations. Overall, our analyses evaluated the phylogenetic identification for *D. eres* species and demonstrated the population diversity of *D. eres* in China.

## 1. Introduction

The genus *Diaporthe*, belonging to the class of Sordariomycetes, the order of Diaporthales, and the family of Diaporthaceae, was originally established with Diaporthe eres Nitschke as the typified species [1,2]. The genus *Diaporthe* (asexual morph, *Phomopsis*) represents a group of cosmopolitan species, including saprophytic, endophytic, and pathogenic ones on different plants [3,4,5,6,7]. Furthermore, *Diaporthe* species were also reported as the causal agents of many important diseases in humans, mammals, and insects [8,9,10,11]. To date, over 1020 names of “*Diaporthe*” and around 950 names of the asexual morph “*Phomopsis*” have been recorded in MycoBank lists (http://www.mycobank.org (accessed on 15 July 2020), of which more than 100 *Diaporthe* and/or *Phomopsis* species have been reported in China [12,13,14,15,16,17,18,19,20,21,22].

*Diaporthe eres* was firstly collected with a type specimen from *Ulmus* sp. in Germany. It was reported that *D. eres* could cause shoot blight on *Acer pseudoplatanus* [23] and *Juglans cinerea* [24]. It is also responsible for umbel browning and stem necrosis on *Daucus carota* [25], leaf necrosis on *Hedera helix* [26], fruit rot on *Vitis* sp. [27], and stem canker and rootstock death on *Malus* spp. [28]. According to recent studies in China, it is responsible for branch canker, leaf blight, and root rot on *Cinnamomum camphora* [29], *Acanthopanax senticosus*, *Castanea mollissima*, *Melia azedarace*, *Rhododendron simsii*, *Sorbus* sp. [20], *Juglans regia* [14,20], *Polygonatum sibiricum* [30], *Photinia fraseri* cv. Red Robin [31], *Coptis chinensis* [32], *Acer palmatum* [33], *Pyrus* sp. [16], *Vitis* sp. [19], *Prunus persica* [13], and *Pinus albicaulis* [34]. It often associates with many important economic trees, e.g., camellia [35,36], camptotheca [37], citrus [18], grapevine [19,38], Japanese oak [39,40], kiwifruit [41], peach [13], pear [16], walnut [14], and so on.

Genealogical Concordance Phylogenetic Species Recognition (GCPSR) [42] represents an enhanced tool for species delimitation in the *Diaporthe* genus compared to morphological and biological identification [14,43]. The species concept in *D. eres* has greatly progressed since the molecular approach of concatenated multigene genealogies under GCPSR started to be conducted. However, the other processes, e.g., incomplete lineage sorting, recombination, and horizontal gene transfer, can cause discordances between gene trees and species trees and mask the true evolutionary relationship among closely related taxa [44]. Furthermore, the regular approach of concatenating sequence data from multiple loci under GCPSR can lead to inconsistency and poor species discrimination [45].

In recent years, insights into the species boundaries of the *Diaporthe* genus have been resolved based on morphological characterization combined with multi-locus phylogenetic analyses [3,4,7,46,47]. Effective multi-locus phylogenetic analyses were employed to identify *Diaporthe* species with ribosomal internal transcribed spacer (ITS), translation elongation factor 1-α gene (*EF1-α*), beta-tubulin gene (*TUB2*), calmodulin gene (*CAL*), and histone-3 gene (*HIS*) [3,15,47,48]. As a result, several *Diaporthe* species with close phylogenetic relevance were successfully demonstrated as synonyms of *D. eres*, including *D. castaneae-mollissimae*, *D. cotoneastri*, *D. nobilis*, *Phomopsis fukushii* [43], *D. biguttusis*, *D. ellipicola*, *D. longicicola*, *D. mahothocarpus* [14], *D. camptothecicola*, and *D. momicola* [20].

Therefore, the objectives of the study are to (i) employ different delimitation methods based on a genomic DNA sequence database to interpret species boundaries and to facilitate further species identification for *D. eres*, (ii) investigate Chinese populations of the *D. eres* species and characterize the relationship between the populations and their distributions based on sequences of multiple loci, and (iii) reconstruct phylogeny and explore the evolution of *D. eres* with the newly updated Chinese population.

## 2. Materials and Methods

### 2.1. D. eres and Related Species Isolates Used

Thirty-seven species, including the *D. eres* species complex and closely related species, were used in phylogenetic analyses. These species were originally collected in Australia, Canada, China, France, Germany, Italy, Japan, Korea, Netherlands, South Africa, Suriname, Thailand, UK, USA, and Yugoslavia, and their corresponding DNA sequences were downloaded from NCBI’s GenBank nucleotide database (www.ncbi.nlm.nih.gov (accessed on 7 April 2020)) (Table 1).

A diversity analysis of *D. eres* populations in China was carried out using 138 isolates that were selected based on different hosts (data from previously published literature), including *Actinidia chinensis* [41], *Camellia* sp. [35,36], *Camptotheca acuminata* [37], *Citrus* spp. [18], *Juglans regia* [14], *Lithocarpus glabra* [39,40], *Prunus persica* [13], *Pyrus* spp. [16], and *Vitis* spp. [19,38]. These isolates were originally collected from not only different hosts but also different areas, including 16 provinces, i.e., Beijing (BJ), Chongqing (CQ), Fujian (FJ), Gansu (GS), Hebei (HEB), Henan (HN), Hubei (HUB), Jiangsu (JS), Jiangxi (JX), Jilin (JL), Liaoning (LN), Ningxia (NX), Shandong (SD), Sichuan (SC), Yunnan (YN), and Zhejiang (ZJ).

### 2.2. Selection of Suitable Markers for Genetic Diversity Analysis

Based on previous studies, ITS, *EF1-α*, *TUB2*, *CAL*, and *HIS* were selected for the evaluation of species diversity of the *Diaporthe* genus in phylogenetic analysis. In brief, the ITS sequence was amplified with the primer set of ITS1/ITS4 [49], *EF1-α* with EF1-728F/EF1-986R [50], *TUB2* with Bt-2a/Bt-2b [51], *CAL* with CAL-228F/CAL-737R [50], and *HIS* with CYLH3F/H3-1b [51,52]. PCR amplification protocols of the five loci are the same as those described previously [53].

### 2.3. Sequence Alignment and Phylogenetic Analyses

Multi-locus phylogenetic analyses were conducted to identify isolates to species level using assembled DNA sequences of five loci. DNA sequences were used for consensus analysis with minor manual editions in the DNASTAR Lasergene Core Suite software program (SaqMan v.7.1.0; DNASTAR Inc., Madison, WI, USA). Sequence alignments and comparisons of assembled sequences were performed using the L-INS-i algorithm on the MAFFT alignment online server v.7.467 [54]. The aligned sequences were checked and manually adjusted in BioEdit v.7.2.5 [55] and converted to suitable formats (PHYLIP and NEXUS) using the Alignment Transformation Environment (ALTER) website online server [56]. The resulting DNA sequences, containing all five loci, were deposited at TreeBASE (submission number: 26697). Maximum likelihood (ML) phylogenetic trees were constructed using RAxML-HPC BlackBox v.8.2.10 [57], available in the CIPRES Science Gateway v.3.3 Web Portal [58], with 1000 bootstrap replications. The general time-reversible model of evolution, including the estimation of invariable sites (GTRGAMMA + I), was performed in ML analysis. Maximum parsimony (MP) analysis was performed with 1000 replicates using Phylogenetic Analyses Using Parsimony (PAUP*) v.4.0b10 [59]. Goodness fit and bootstrap values were calculated and harvested from tree length (TL), the consistency index (CI), the retention index (RI), the rescaled consistency index (RC), and the homoplasy index (HI). A heuristic search was carried out with 1000 random stepwise addition replicates using the tree bisection-reconnection (TBR) branch-swapping algorithm on “best trees”. Gaps were treated as missing data, and all characters were weighted equally. The bootstrap support values (BS) were determined by the software to assess the robustness of MLBS and MPBS analyses; only branches with MLBS and MPBS over 70% were considered for ML/MP phylogenetic inference. Posterior probabilities values (PP) were calculated by Markov Chain Monte Carlo (MCMC) sampling in MrBayes v.3.2.2 [60]. The best-fit model of nucleotide substitution was determined with corrected Akaike information criterion (AIC) in MrModeltest v.2.3 [61] (Appendix A). For BI analysis, four MCMC chains were run simultaneously, starting from random trees, for 10^5^ generations, and trees were sampled every 100^th^ generation. The calculation of BI analysis was stopped when the average standard deviation of split frequencies fell below 0.01. The first 10% of resulting BI trees, which represent the burn-in phase of the analysis by inspecting likelihoods and parameters in Tracer v.1.7.1 [62], were discarded, and the remaining 9000 trees were used to calculate the posterior probabilities (PP) in the majority rule consensus tree. Bayesian posterior probability values (BIPP) over 0.95 were considered for BI trees, and all trees were rooted with *D. citri* (CBS 135422).

### 2.4. Genealogical Concordance Phylogenetic Species Recognition (GCPSR) Analysis

In this study, species boundaries were determined using genealogical concordance phylogenetic species recognition (GCPSR), as described in previous studies, in SplitsTree4 v.4.14.6 (www.splitstree.org (accessed on 26 September 2017)) [42,63,64]. Multi-locus concatenated sequence data, with *EF1-α*, *TUB2*, *CAL*, and *HIS,* were used to determine the recombination level within phylogenetically closely related species. In addition, the results of relationships between closely related species were visualized by constructing neighbor-joining (NJ) graphs.

### 2.5. Phylogenetic Informativeness Analysis

Phylogenetic informativeness (PI) was analyzed from taxonomically authenticated species and type-strains based on the multi-locus combined dataset of ITS, *EF1-α*, *TUB2*, *CAL*, and *HIS*. Twenty-eight representative isolates (from 23 species, including an outgroup) with a close relationship to the *D. eres* species complex based on phylogenetic analysis were selected to determine the profiling of phylogenetic informativeness [65]. Ultrametric trees were generated from the concatenated alignment dataset using maximum likelihood (ML) phylogenetic analysis, as described above. To estimate phylogenetic informativeness (phylogenetic informativeness per site (PI per site) and net phylogenetic informativeness (Net PI)), the corresponding partitioned alignment was harvested from the PhyDesign Web Portal at http://phydesign.townsend.yale.edu/ (accessed on 22 April 2020) [66].

### 2.6. Population Aggregation and Haplotype Network Analysis

To confirm the *D. eres* species, 138 taxa (Appendix A), along with 51 taxa (Table 1), were reconstructed using multi-locus sequences of *EF1-α*, *TUB2*, *CAL*, and *HIS* (Appendix A). In order to analyze the genetic diversity for *D. eres* populations, isolates that have been analyzed in phylogenetic analyses were applied. In brief, an individual locus was sequenced, and the alignment and comparison of assembled sequences were performed using ClustalX v.2.0.11 [67]. Gaps were treated as the missing data of each locus, and the end of 5′- and 3′- partial sequences were trimmed in the dataset. All population genetic parameters, including the number of polymorphic (segregating) sites (S), Nei’s nucleotide diversity (π), haplotype numbers (Hap), haplotype diversity (Hd), nucleotide diversity from S (θw), and neutrality statistic information, such as Tajima’s *D*, Fu and Li’s *D*, and Fu’s *Fs,* were calculated using DnaSP v.6.11.01 [68] for each individual locus and combined loci. Therefore, relationships among the haplotypes were depicted with the median-joining (MJ) method in Population Analysis with Reticulate Trees (PopART: http://popart.otago.ac.nz/index.shtml (accessed on 1 June 2020)) [69].

**Table 1 biology-10-00179-t001:** List of *D. eres* species complex isolates used for phylogenetic analyses, with details of host, origin, and GenBank accession number.

*Diaporthe* Species ^a^	Isolate Number ^b^	Origin	GenBank Accession Numbers ^c^					**References**
			ITS	*EF1-α*	*TUB2*	*CAL*	*HIS*	
*D. acerigena* ^T^	**CFCC 52554**	China	MH121489	MH121531	–	MH121413	MH121449	[20]
*D. alleghaniensis* ^T^	**CBS 495.72**	Canada	KC343007	KC343733	KC343975	KC343249	KC343491	[46]
*D. alnea*	**CBS 146.46**	Netherlands	KC343008	KC343734	KC343976	KC343250	KC343492	[46]
*D. apiculatum* ^T^	**CGMCC3.17533**	China	KP267896	KP267970	KP293476	–	–	[15]
*D. betulae* ^T^	**CFCC 50469**	China	KT732950	KT733016	KT733020	KT732997	KT732999	[70]
*D. betulina* ^T^	**CFCC 52562**	China	MH121497	MH121539	MH121579	MH121421	MH121457	[20]
*D. bicincta* ^EP^	**CBS 121004**	USA	KC343134	KC343860	KC344102	KC343376	KC343618	[43,46]
*D. celastrina* ^EP^	**CBS 139.27**	USA	KC343047	KC343773	KC344015	KC343289	KC343531	[43,46]
*D. celeris* ^T^	**CBS 143349**	UK	MG281017	MG281538	MG281190	MG281712	MG281363	[4]
*D. charlesworthii* ^T^	**BRIP 54884m**	Australia	KJ197288	KJ197250	KJ197268	–	–	[6]
*D. chensiensis* ^T^	**CFCC 52567**	China	MH121502	MH121544	MH121584	MH121426	MH121462	[20]
*D. citri* ^T^	**CBS 135422**	USA	KC843311	KC843187	KC843071	KC843157	MF418281	[3,7]
*D. citrichinensis* ^T^	**CGMCC3.15225**	China	JQ954648	JQ954666	MF418524	KC357494	KJ490516	[3,17,18]
*D. citrichinensis*	**ZJUD034B**	China	KJ210539	KJ210562	KJ420829	KJ435042	KJ420879	[43]
*D. collariana* ^T^	**MFLUCC 17-2636**	Thailand	MG806115	MG783040	MG783041	MG783042	–	[71]
*D. conica* ^T^	**CFCC 52571**	China	MH121506	MH121548	MH121588	MH121428	MH121466	[20]
*D. eres* ^EP^	**CBS 138594**	Germany	KJ210529	KJ210550	KJ420799	KJ434999	KJ420850	[43]
*D. eres* (*D. biguttusis*) ^T^	**CGMCC3.17081**	Unknown	KF576282	KF576257	KF576306	–	–	[39]
*D. eres* (*D. camptothecicola*) ^T^	**CFCC 51632**	China	KY203726	KY228887	KY228893	KY228877	KY228881	[37]
*D. eres* (*D. castaneae-mollissimae*) ^T^	**DNP128**	China	JF957786	KJ210561	KJ420801	KJ435040	KJ420852	[43,72]
*D. eres* (*D. cotoneastri*) ^T^	**CBS 439.82**	UK	FJ889450	GQ250341	JX275437	JX197429	–	[72]
*D. eres* (*D. ellipicola*) ^T^	**CGMCC3.17084**	China	KF576270	KF576245	KF576291	–	–	[39]
*D. eres* (*D. henanensis*) ^T^	**CGMCC3.17639**	China	KC898258	–	KF600608	–	KF600609	[73]
*D. eres* (*D. longicicola*) ^T^	**CGMCC3.17089**	Unknown	KF576267	KF576242	KF576291	–	–	[39]
*D. eres* (*D. lonicerae*) ^T^	**MFLUCC 17-0963**	Italy	KY964190	KY964146	KY964073	KY964116	–	[74]
*D. eres* (*D. mahothocarpus*) ^T^	**CGMCC3.15181**	China	KC153096	KC153087	KF576312	–	–	[39,40]
*D. eres* (*D. momicola*) ^T^	**CGMCC3.17466**	China	KU557563	KU557631	KU557587	KU557611	–	[13]
*D. eres* (*D. nobilis)*	**CBS 113470**	Korea	KC343146	KC343872	KC344114	KC343388	KC343630	[46]
*D. eres* (*D. rosicola*) ^T^	**MFLU 17-0646**	UK	MG828895	MG829270	MG843877	–	–	[75]
*D. eres* (*Phomopsis fukushii*) ^NE^	**MAFF 625033**	Japan	JQ807468	JQ807417	KJ420814	KJ435017	KJ420865	[43]
*D. eucommiicola* ^H^	**SCHM 3607**	China	AY578071	–	–	–	–	[76]
*D. fraxinicola* ^T^	**CFCC 52582**	China	MH121517	MH121559	–	MH121435	–	[20]
*D. gardeniae*	**CBS 288.56**	Italy	KC343113	KC343839	KC344081	KC343355	KC343597	[46]
*D. helicis* ^EP^	**CBS 138596**	France	KJ210538	KJ210559	KJ420828	KJ435043	KJ420875	[43]
*D. heterophyllae* ^T^	**CBS 143769**	France	MG600222	MG600224	MG600226	MG600218	MG600220	[77]
*D. infertilis* ^T^	**CBS 230.52**	Suriname	KC343052	KC343778	KC344020	KC343294	KC343536	[3,46]
*D. maritima* ^T^	**DAOMC 250563**	Canada	KU552025	KU552023	KU574615	–	–	[78]
*D. neilliae*	**CBS 144.27**	Unknown	KC343144	KC343870	KC344112	KC343386	KC343628	[46]
*D. oraccinii* ^T^	**CGMCC3.17531**	China	KP267863	KP267937	KP293443	–	KP293517	[15]
*D. padina* ^T^	**CFCC 52590**	China	MH121525	MH121567	MH121604	MH121443	MH121483	[20]
*D. penetriteum* ^T^	**CGMCC3.17532**	China	KP714505	KP714517	KP714529	–	KP714493	[15]
*D. phragmitis* ^T^	**CBS 138897**	China	KP004445	–	KP004507	–	KP004503	[79]
*D. pulla*	**CBS 338.89**	Yugoslavia	KC343152	KC343878	KC344120	KC343394	KC343636	[43]
*D. sambucusii* ^T^	**CFCC 51986**	China	KY852495	KY852507	KY852511	KY852499	KY852503	[80]
*D. sennicola* ^T^	**CFCC 51634**	China	KY203722	KY228883	KY228889	KY228873	–	[81]
*D. shennongjiaensis* ^T^	**CNUCC 201905**	China	MN216229	MN224672	MN227012	MN224551	MN224559	[35]
*D. subclavata* ^T^	**CGMCC3.17257**	China	KJ490630	KJ490509	KJ490451	–	KJ490572	[18]
*D. tibetensis* ^T^	**CFCC 51999**	China	MF279843	MF279858	MF279873	MF279888	MF279828	[14]
*D. ukurunduensis* ^T^	**CFCC 52592**	China	MH121527	MH121569	–	MH121445	MH121485	[20]
*D. vaccinii* ^T^	**CBS 160.32**	USA	KC343228	KC343954	KC344196	KC343470	KC343712	[46]
*D. virgiliae* ^T^	**CBS 138788**	South Africa	KP247573	–	KP247582	–	–	[82]

^a^ H (holotype), ^T^ (ex-type), ^EP^ (ex-epitype), and ^NE^ (ex-neotype) cultures are indicated with isolate numbers in bold. ^b^ BRIP: Plant Pathology Herbarium, Department of Employment, Economic, Development and Innovation, Queensland, Australia; CBS: Westerdijk Fungal Biodiversity Institute, Utrecht, The Netherlands; CFCC: China Forestry Culture Collection Center, Beijing, China; CGMCC: China General Microbiological Culture Collection, China; CNUCC: Capital Normal University Culture Collection Center, Beijing, China; DAOMC: Canadian Collection of Fungal Cultures, Agriculture, and Agri-Food Canada, Ottawa, Canada; DNP: First author’s personal collection (deposited in MFLUCC), Thailand; MAFF: NIAS GenBank Project, Ministry of Agriculture, Forestry, and Fisheries, Japan; MFLU: Herbarium of Mae Fah Luang University, Chiang Rai, Thailand; MFLUCC: Mae Fah Luang University Culture Collection, Chiang Rai, Thailand; SCHM: Mycological Herbarium of South China Agricultural University, Guangzhou, China; ZJUD: *Diaporthe* species culture collection at the Institute of Biotechnology, Zhejiang University, Hangzhou, China. ^c^ Ribosomal internal transcribed spacer (ITS) region of ribosomal DNA (ITS1-5.8S-ITS2), translation elongation factor 1-α (*EF1-α*), beta-tubulin 2 (*TUB2*), calmodulin (*CAL*), and histone-3 (*HIS*).

## 3. Results

### 3.1. Phylogenetic Analysis of D. eres

In this study, the concatenated DNA sequences of five loci (ITS, *EF1-α*, *CAL*, *TUB2*, and *HIS*) from 216 sequences, including 5 outgroup sequences of *D. citri*, were used to infer delimitation of *Diaporthe* species. For the reconstruction of phylogenetic trees of *Diaporthe* species, altogether, 51 sequences of ITS, 47 sequences of *EF1-α*, 36 sequences of *CAL*, 47 sequences of *TUB2*, and 35 sequences of *HIS* were obtained from the GenBank database. Sequences of ITS, *EF1-α*, *CAL*, *TUB2*, and *HIS* were determined as 598, 592, 542, 828, and 502 base pairs (bp), respectively. For species delimitation of the *D. eres* complex, 50 taxa were analyzed, with 2464 bp assembled sequences of 4 genes, including 592 bp (1–592) of *EF1-α*, 542 bp (593–1134) of *CAL*, 828 bp (1135–1962) of *TUB2*, and 502 bp (1963–2464) of *HIS*, respectively. For the 5 loci combined sequences dataset with ITS region, we filled in the end of the four-gene dataset with 598 bp (2465–3062) of ITS. ML, MP, and BI analyses were used to perform phylogenetic reconstruction for individual and combined datasets; results showed similar topology and few differences in statistical support values. A comparison of alignment properties in parsimony analyses of individual and combined loci used in phylogenetic analyses is provided in Table 2.

Phylogenetic analyses for *Diaporthe* species were performed using each individual locus and combined loci of DNA sequences (Appendix A). Among them, phylogenetic trees, using *EF1-α*, *EF1-α+CAL*, and *EF1-α+CAL+TUB2*+*HIS*, showed clear delimitation for *D. eres* species, while unclear delimitation was observed using five loci sequences of *EF1-α+CAL+TUB2*+*HIS*+ITS (Figure 1). It was found that the three-loci combined dataset of *EF1-α+CAL+TUB2* or *EF1-α+CAL*+*HIS*, with ML, MP, and BI analyses, was unable to separate *D. bicincta* (CBS 121004), *D. celastrina* (CBS 139.27), *D. celeris* (CBS 143349), *D. helicis* (CBS 138596), *D. maritima* (DAOMC 250563), *D. phragmitis* (CBS 138897), and *D. pulla* (CBS 338.89) species (Appendix A). Overall, the four-loci combined dataset of *EF1-α+CAL+TUB2*+*HIS* showed the highest reliability to identify and resolve species boundaries in the *D. eres* complex (Figure 1 and Figure 2).

The Bayesian inference phylogenetic tree of the *D. eres* species complex and close species constructed with combined DNA sequences is presented in Figure 2 as an example. In the phylogenetic tree, the *D. eres* species complex cluster (*D. eres* species complex) was clearly separated from another cluster that included *D. collariana*, *D. heterophyllae*, *D. virgiliae*, *D. penetriteum*, *D. infertilis*, *D. sambucusii*, and *D. shennongjiaensis.* Within the *D. eres* species complex cluster, a subcluster that included *D. eres* and 13 species with other names should be the synonymous species of *D. eres*. The close subcluster, which included *D. helicis* (CBS 138596), *D. pulla* (CBS 338.89), *D. phragmitis* (CBS 138897), and *D. celeris* (CBS 143349), showed a relatively distinct distance with *D. eres*, indicating that they are different species (Figure 2A). The *D. eres* species was further analyzed by GCPSR analysis. The NJ tree shows the relationship between Cluster I and Cluster II (Figure 2B), indicating that genetic diversity is rich in this species.

### 3.2. Phylogenetic Informative Analysis

For phylogenetic informative analysis, only taxa with complete 5 loci sequences were used. As a result, the assembled DNA sequences were 3049 bp, including 544 bp of *CAL* (1–544), 573 bp of *EF1-α* (545–1117), 829 bp of *TUB2* (1118–1946), 503 bp of *HIS* (1947–2449), and 600 bp of ITS (2450–3049). The combined dataset consisted of 28 taxa (from 23 species), including the outgroup species *D. citri* (CBS 135422).

Phylogenetic informativeness (PI) profiles, both Net PI and PI per site, indicated that *EF1-α* and *CAL* displayed the highest informative sequences to resolve the phylogenetic signal at the taxonomic level. Next were *HIS* and *TUB2*, which maintained fairly high informative sequences (Figure 3 and Appendix A). ITS presented the lowest PI signal among the selected loci and was unreliable for the delimitation of the *D. eres* species. The combined dataset of four loci (*EF1-α*, *TUB2*, *CAL*, and *HIS*) showed better delimitation for *D. eres* compared to the dataset of five loci (Figure 3), further confirming that the ITS locus was lowly informative.

### 3.3. D. eres Species Boundaries

Based on the phylogenetic analyses using multi-locus reconstruction (*EF1-α*, *TUB2*, *CAL*, and *HIS*), the species delimitation was determined among *D. eres* and closely related species (Figure 1 and Figure 2). Results showed that *D. eres* and the other 13 species were conspecific. Among them, *D. biguttusis*, *D. camptothecicola*, *D. castaneae-mollissimae*, *D. cotoneastri*, *D. ellipicola*, *D. longicicola*, *D. mahothocarpus*, *D. momicola*, *D. nobilis*, and *Phomopsis fukushii* have already been previously considered the synonymous species of *D. eres*; in this study, another three species, *D. henanensis*, *D. lonicerae*, and *D. rosicola*, were further revealed to be synonyms of *D. eres*.

*Diaporthe eres* Nitschke 1870 [83].

= *Diaporthe henanensis* Y. Yang, H.Y. Wu & M. Zhang, 2016.

= *Diaporthe lonicerae* A.J. Dissanayake, E. Camporesi & K.D. Hyde, 2017.

= *Diaporthe rosicola* D.N. Wanasinghe, E.B.G. Jones & K.D. Hyde, 2018.

The detailed description and illustrations for these species can be found in previous reports [73,74,75].

### 3.4. Population Aggregation and Haplotype Network Analysis

The ITS of 137 taxa, *EF1-α* of 132 taxa, *TUB2* of 137 taxa, *CAL* of 118 taxa, *HIS* of 70 taxa, and the combined sequences of 61 taxa were 450, 278, 355, 492, 421, 1429 (four loci without ITS), and 1882 (five loci) bp in length, respectively. The analysis of genetic diversity within *D. eres* showed a high level of haplotype numeric (Hap) and haplotype diversity (Hd). The summary of sequence variation and indices of sequence variation within the five loci among *D. eres* are shown in Table 3 and Appendix A. The haplotype diversity values of ITS, *EF1-α*, *TUB2*, *CAL*, *HIS*, and the combined datasets were greater than 0.5, reflecting high genetic diversity. The neutrality statistic (Tajima’s *D* and Fu’s *Fs*) results showed negative values, suggesting population expansion in *D. eres* isolates. We obtained similar results of population network analysis in the phylogenetic tree (Figure 4A) and the median-joining haplotype network (Figure 4B) using the combined dataset of *EF1-α*, *TUB2*, *CAL*, and *HIS*. Population connectivity was grouped into two clusters that were not correlated to specific populations of geographic distribution (Figure 5).

### 3.5. Phylogenetic Informative Analysis

It should be noted that median-joining haplotype network analysis was also performed based on each individual locus using DnaSP v.6.11.01. The major haplotype numbers from *EF1-α*, *CAL*, *TUB2*, *HIS*, and ITS were 29, 49, 25, 23, and 9, respectively. Analysis with the CAL locus showed two small distinct clusters: one consisted of hap 1, 18, and 21 from BJ, GS, HN, HUB, JL, LN, and SD, and another consisted of hap 5, 9, and 10 from CQ, HUB, JX, and YN. HUB isolates could be found in both clusters (Figure 5). Similarly, the analysis of the *TUB2* locus also showed two small clusters. Thus, we found that haplotypes that were connected between Cluster I (hap 11) and Cluster II (hap 13, 21, and 26) were from a center part of China, i.e., HEB, HN, HUB, and JX. Analysis of *EF1-α*, *HIS*, and ITS loci showed a wide distribution and incommensurate derivative splitting by geographic distribution. These median-joining haplotype networks in each individual locus are shown in Appendix A.

## 4. Discussion

In this study, we used five-loci DNA sequences to understand and interpret the species boundaries of the *D. eres* species complex and assess the genetic diversity of *D. eres* populations in China. A total of 51 taxa, including 37 close species to the *D. eres* species complex, was applied to narrow the criteria of phylogenetic relatives using the GCPSR of phylogenetic species, while 138 *D. eres* isolates from various Chinese populations were examined to assess the relationship between genetic diversity and different geographic distributions.

Recently, the classification of *Diaporthe* species has become more dependent on a molecular approach rather than traditional morphological characterization [72,84,85]. Next-generation sequencing (NGS) technology, such as DNA barcoding, is highly efficient, more accurate, and, thus, valid for fungal identification at the species level [86,87]. The ITS sequence is commonly used for preliminary fungal identification and is recommended for identifying species boundaries in the genus *Diaporthe*, *Diaporthaceae*, and Sordariomycetes [5,77,88,89]. However, there are many intraspecific variations in the ITS locus of certain *Diaporthe* species. Sometimes the intraspecific variation is even greater than the interspecific variation, which makes it difficult to identify *Diaporthe* species using the ITS sequence alone [90,91].

The identification of *Diaporthe* species based on morphological characterization is very contradictory, and a molecular approach using DNA sequences should be combined to identify species within this genus [46,47]. To redefine the boundaries of *Diaporthe* species, Santos et al. [47] proposed highly effective phylogenetic reconstruction using DNA barcoding sequences of multiple loci, i.e., ITS, *EF1-α*, *TUB2*, *CAL*, and *HIS*. The taxonomy of *Diaporthe* is complex, and many *Diaporthe* spp. are classified based on different criteria, according to host associations, morphological characteristics [12,92,93,94], or sequences of the ITS region [5,92,95]. It is suggested that only the type strains whose identification has been widely recognized should be accepted as references for the taxonomy of this genus [46,96,97]. In this study, several isolates, including type strains from previous publications, were selected as references with phylogenetic analysis. However, when a MegaBlast search was performed for each locus in NCBI, generally, the *Diaporthe* species showing the highest similarity with the sequence of each locus of the isolates were not the type strains. Thus, the species used by us in the current study were not always the same as those recovered by the single locus MegaBlast search in NCBI. The combined multi-locus phylogenetic reconstruction shows the very strong species delimitation for the *D. sojae* complex [48]. Fan et al. [14] demonstrated the effectiveness of 3 loci, including *EF1-α*, *TUB2*, and *CAL*, for the identification of the *D. eres* complex in walnut trees. Similarly, Yang et al. [20] and Zhou and Hou [35] also used three-locus sequences to identify *D. eres* species associated with different hosts in China. These studies excluded a few closely related *Diaporthe* species with typical reference strains. However, our study revealed that the phylogenetic analysis from the combined dataset of *EF1-α*, *TUB2*, *CAL*, and *HIS* was highly effective and strongly supported to resolve species boundaries of the *D. eres* species complex. This is consistent with the results obtained by Guo et al. [16]. Phylogenetic informative (PI) profiles using multi-locus phylogenetic analysis of five loci are commonly used to identify the *D. eres* species complex; both Net PI and PI per site showed similar results. Among different loci, *EF1-α*, *APN2* (DNA lyase), and *HIS* loci are effective for the species delimitation of the *D. eres* species complex. It was reported that *EF1-α* showed the highest effectiveness to resolve the phylogenetic signal, which is concordant with the results obtained by Udayanga et al. [43]. Similarly, the highly variable *EF1-α* locus showed the highest effectiveness to discriminate species in the *Diaporthe* genus [43,47,91,98]. Our study revealed that *EF1-α* was reliable, but the ITS region impeded species delimitation and relatively limited phylogenetic signals when the combined DNA sequences of five loci (ITS, *EF1-α*, *TUB2*, *CAL*, and *HIS*) were used.

In previous studies, several synonyms of the *D. eres* species were successfully demonstrated based on phylogenetic analyses using multi-locus sequences, i.e., *D. biguttusis*, *D. camptothecicola*, *D. castaneae-mollissimae*, *D. cotoneastri*, *D. ellipicola*, *D. longicicola*, *D. mahothocarpus*, *D. momicola*, *D. nobilis*, and *Phomopsis fukushii* [14,20,43]. In the current study, we found that *D. henanensis* [73], *D. lonicerae* [74], and *D. rosicola* [75] were also synonyms of *D. eres* because these 3 species and the previously demonstrated 10 species were grouped into a single subcluster with *D. eres* in phylogenetic analyses.

Using population genetic analyses, Manawasighe et al. [19] demonstrated the genetic variation of *D. eres* associated with grapevine dieback in China and found isolates grouped according to geographic location. However, a comparison of Chinese and European *D. eres* isolates, using both individual- and multi-locus DNA sequences of ITS, *EF1-α*, *TUB2*, *CAL*, and *HIS* loci, did not show significant differences between the two geographical populations. In this study, *D. eres* were grouped into two major populations that were not correlated with geographic distribution. Interestingly, we found that isolates from the central part of China, e.g., HEB, HN, HUB, and JX, simultaneously fell into two different clusters with a significant haplotype connection, suggesting that this region is the origin of *D. eres*. This is consistent with the observation that HUB isolates might be the parental population of *D. eres* [19].

Finally, future species identification should use a highly effective molecular approach to make it simple and easy to detect *D. eres* in routine plant quarantine. For genetic variation and population analyses, sample sizes should be increased and comparisons should be performed with the analyses using other molecular markers, including amplified fragment length polymorphism (AFLP), random amplified polymorphic DNA (RAPD), and inter simple sequence repeat (ISSR). Further understanding should focus on the ancestor, phylogeographic and demographic history, divergence-time estimation, and migration history of *D. eres*.

## 5. Conclusions

The current study provides an overview of *D. eres* on several plant varieties and some valuable knowledge to identify this fungus. Phylogenetic trees were reconstructed for the *D. eres* species complex with combined DNA sequences of *EF1-α*, *CAL*, *TUB2*, and *HIS*. Phylogenetic analyses and phylogenetic informativeness profiles reported in this study revealed that for *D. eres* species identification and delimitation, the usage of the *EF1-α* locus represents the optimal alternative; this proposition is also supported by previous studies [43,47,91,98]. Moreover, our analyses revealed that the usage of the ITS region hampers proper species recognition within the *D. eres* species complex. An expansion of population connectivity among the *D. eres* populations was detected. One hundred and thirty-eight *D. eres* isolates were divided by phylogenetic analyses, and genetic distance estimation with haplotype networks revealed two clusters with strong support from population genetic parameters and neutrality of statistic informative values, indicating a high level of haplotype diversity. However, we found that the two clusters from both methods were not separated based on geographic distribution. Overall, our analyses determined the current pattern of phylogenetic identification for the *D. eres* species and population diversity within the Chinese isolates of *D. eres*. In the future, studies on the evolution of *D. eres* and plant-*D. eres* interaction should be conducted.

## Figures and Tables

**Figure 1 biology-10-00179-f001:**
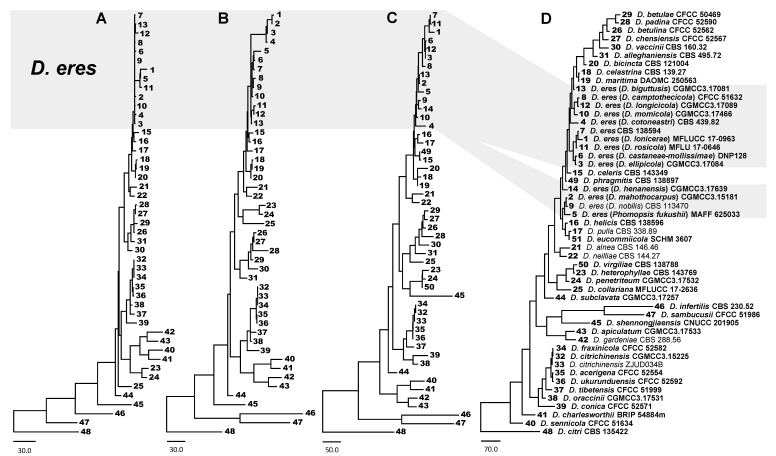
The best parsimonious trees obtained from a heuristic search for *D. eres* and closely related species. The tree was rooted using *D. citri* (CBS 135422). (**A**) *EF1-α* locus. (**B**) Combined dataset of 2 loci (*EF1-α*+*CAL*). (**C**) Combined dataset of 4 loci (*EF1-α*+*CAL*+*TUB2*+*HIS*). (**D**) Combined dataset of 5 loci (*EF1-α*+*CAL*+ *TUB2*+*HIS*+ITS). Taxa numbers were generated and corresponded to samples in 5 multi-locus parsimonious trees. Holotype, ex-type, ex-epitype, and ex-neotype cultures are indicated with isolate numbers in bold.

**Figure 2 biology-10-00179-f002:**
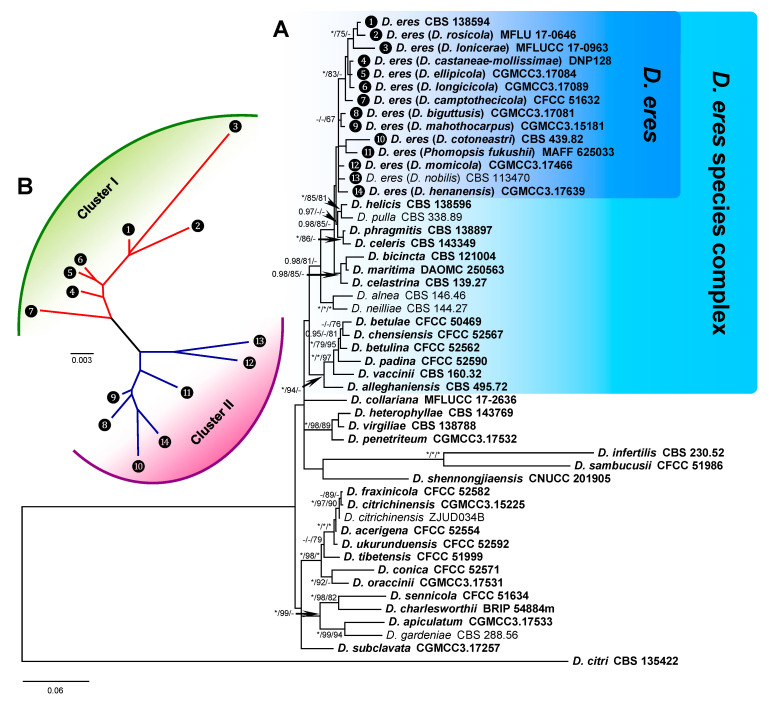
Phylogenetic tree of the *D**. eres* species complex and close species inferred from a combined alignment with a multi-locus dataset (*EF1-α*+*CAL*+*TUB2*+*HIS*). (**A**) The majority-rule consensus tree from Bayesian inference analysis showing the phylogenetic relationships between the *D. eres* species complex and close species. The tree was rooted using *D. citri* (CBS 135422). Bayesian posterior probabilities values (BIPP) >0.95 and maximum likelihood and maximum parsimony bootstrap values (MLBS and MPBS) >70% are given at the branch nodes (BIPP/MLBS/MPBS). Fully supported branched values, with BIPP = 1.0, MLBS and MPBS = 100, are indicated with an asterisk (*). Holotype, ex-type, ex-epitype, and ex-neotype cultures are indicated with isolate numbers in bold. (**B**) Unrooted tree of *D. eres* species based on multi-locus sequences. The splits graphs were obtained using both LogDet transformation and neighbor-joining (NJ) distance transformation. Data are from the subset of 14 representative samples that were generated and corresponded to the BI phylogenetic tree.

**Figure 3 biology-10-00179-f003:**
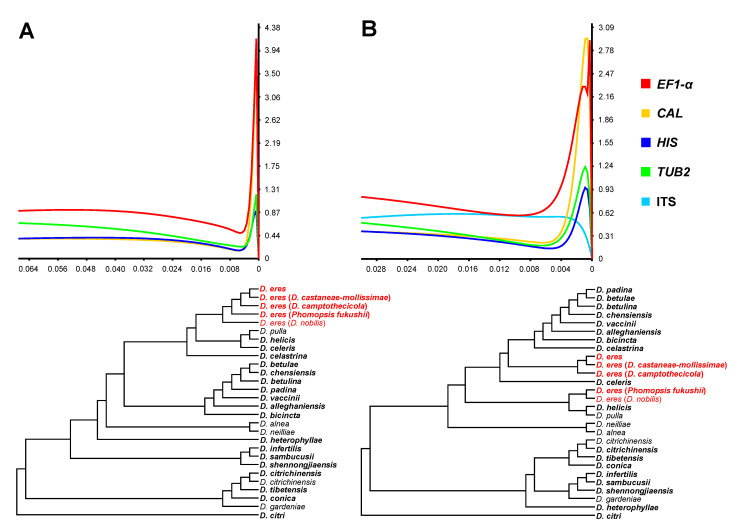
Phylogenetic informativeness profile and ultrametric trees of markers used for phylogenetic studies of 23 species in the *D. eres* species complex and close species. (**A**) Combined dataset from four loci (*EF1-α*+*CAL*+*TUB2*+*HIS*). (**B**) Combined dataset from five loci (*EF1-α*+*CAL*+*TUB2*+*HIS*+ITS). Values on the X-axes correspond to the relative timescale (0–1). Values on the Y-axes represent Net PI (10^3^) in arbitrary units.

**Figure 4 biology-10-00179-f004:**
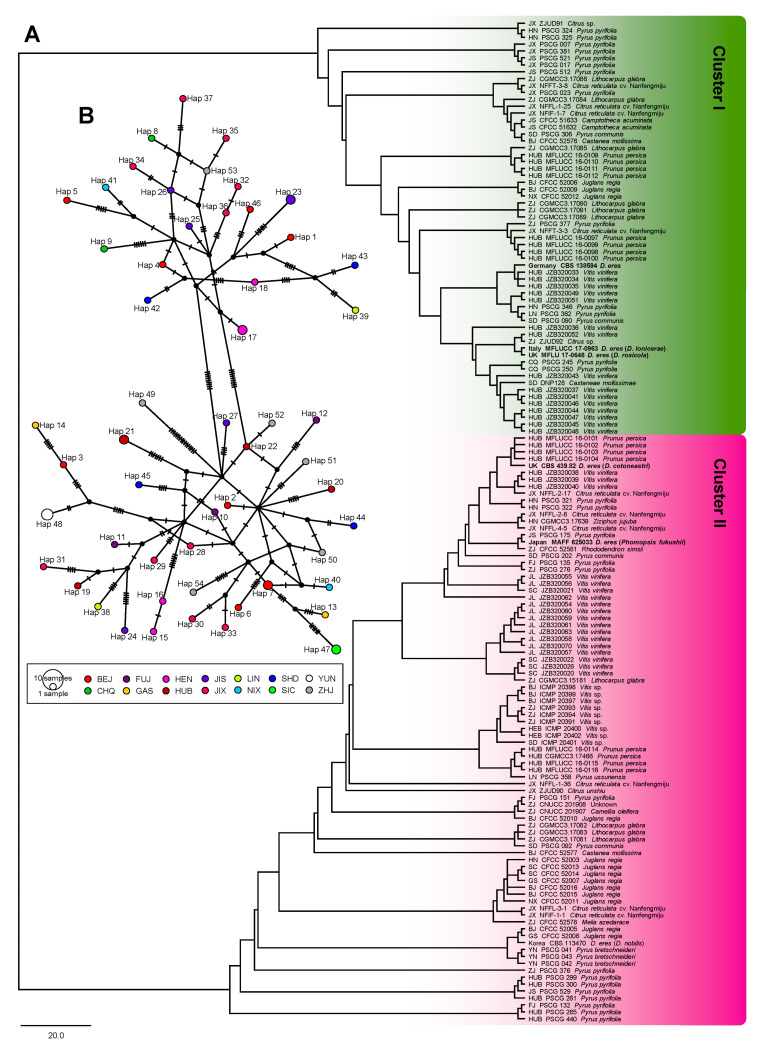
Subdivisions within the *D. eres* population were estimated by using genetic distance generated from the combined dataset of four loci (*EF1-α*+*CAL*+*TUB2*+*HIS*). (**A**) Diversification dynamics from the phylogenetic tree with the neighbor-joining (NJ) method. (**B**) Median-joining (MJ) haplotype network. Each circle represents a unique haplotype, and its size reflects the number of individuals expressing that haplotype. Crosshatches indicate the number of nucleotide differences between haplotypes. Color codes denote the geographic location of populations. Geographic location abbreviation, BJ: Beijing; CQ: Chongqing; FJ: Fujian; GS: Gansu; HEB: Hebei; HN: Henan; HUB: Hubei; JS: Jiangsu; JX: Jiangxi; JL: Jilin; LN: Liaoning; NX: Ningxia; SD: Shandong; SC: Sichuan; YN: Yunnan; ZJ: Zhejiang.

**Figure 5 biology-10-00179-f005:**
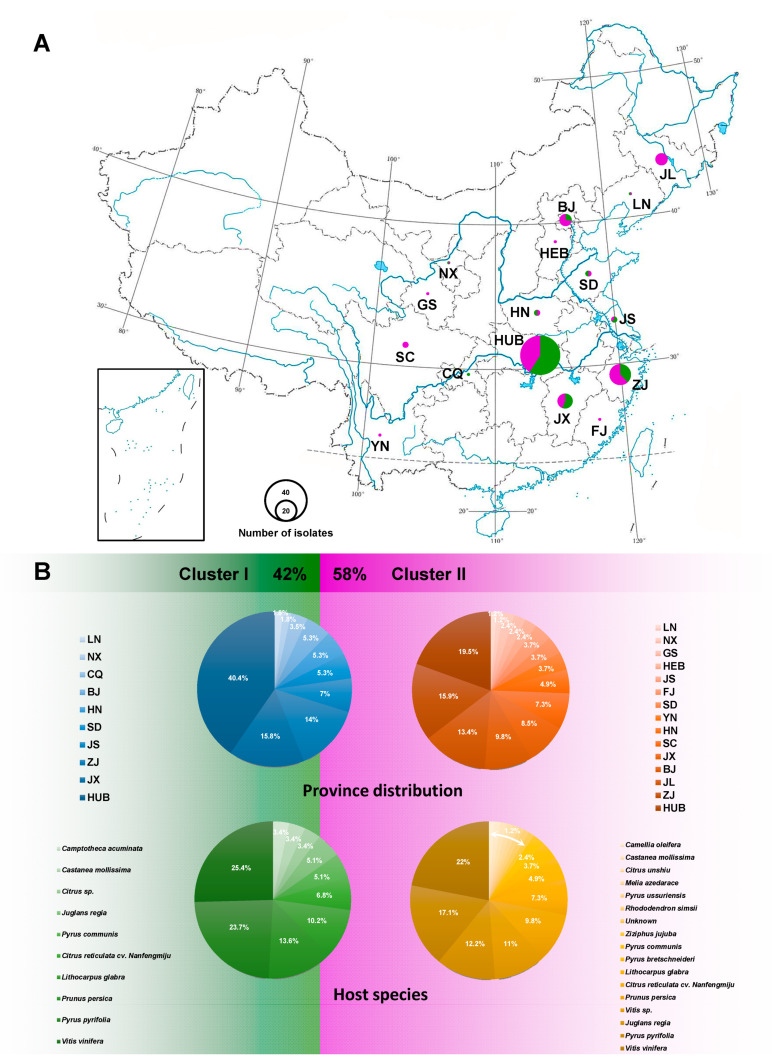
The population prevalence of *D. eres* isolates. (**A**) Province distribution of *D. eres* in China; the size of the circle indicates the number of isolates collected from that location; green and purple colors represent Clusters I and II, respectively. (**B**) Overall *D. eres* population rate (%) of two clusters displayed within province distribution and host species, respectively. Geographic location abbreviation, BJ: Beijing; CQ: Chongqing; FJ: Fujian; GS: Gansu; HEB: Hebei; HN: Henan; HUB: Hubei; JS: Jiangsu; JX: Jiangxi; JL: Jilin; LN: Liaoning; NX: Ningxia; SD: Shandong; SC: Sichuan; YN: Yunnan; ZJ: Zhejiang.

**Table 2 biology-10-00179-t002:** Comparison of alignment properties in phylogenetic data of each individual locus and combined loci.

Locus ^a^	Individual Locus					Combined Loci									
	1	2	3	4	5	1 + 2	1 + 3	1 + 4	2 + 3	2 + 4	3 + 4	1 + 2 + 3	1 + 2 + 4	1 + 2 + 3 + 4	1 + 2 + 3 + 4 + 5
No. of taxa analyzed	47	36	47	35	51	47	50	49	50	41	49	50	49	50	51
Aligned length (with gap)	592	542	828	502	598	1134	1420	1094	1370	1044	1330	1962	1636	2464	3062
Invariable characters	289	319	449	355	466	608	738	644	768	674	804	1057	963	1412	1878
Number of parsimony-informative characters	171	105	136	83	74	276	307	254	241	188	219	412	359	495	569
Number of parsimony-uninformative characters	132	118	243	64	58	250	375	196	361	674	307	493	314	557	615
Tree length (TL)	746	371	597	281	348	1150	1379	1072	995	670	918	1786	1474	2124	2592
Consistency index (CI)	0.635	0.801	0.782	0.698	0.511	0.670	0.682	0.625	0.768	0.736	0.722	0.693	0.656	0.675	0.622
Retention index (RI)	0.698	0.793	0.712	0.767	0.739	0.699	0.676	0.681	0.714	0.755	0.688	0.680	0.688	0.667	0.640
Rescaled consistency index (RC)	0.444	0.635	0.557	0.535	0.378	0.468	0.461	0.426	0.549	0.555	0.497	0.471	0.451	0.451	0.398
Homoplasy index (HI)	0.365	0.199	0.218	0.302	0.489	0.330	0.318	0.375	0.232	0.264	0.278	0.307	0.344	0.325	0.378

^a^ 1: translation elongation factor 1-α gene (*EF1-α*); 2: calmodulin gene (*CAL*); 3: beta-tubulin 2 gene (*TUB2*); 4: histone-3 gene (*HIS*); 5: ribosomal internal transcribed spacer (ITS) region of ribosomal DNA (ITS1-5.8S-ITS2).

**Table 3 biology-10-00179-t003:** Sequence variation, indices of sequence variation, and neutrality within five loci in *D. eres*.

Gene/Locus ^a,b^	Individual Locus					Combined Loci
	1	2	3	4	5	1 + 2 + 3 + 4
Aligned length (with gap)	278	492	355	421	450	1429
No. of taxa analyzed	132	118	137	70	137	61
No. of sites	382	323	794	479	596	1464
%GC	0.548	0.56	0.567	0.62	0.53	0.579
No. of polymorphic (segregating) sites (S)	130	27	58	34	83	115
Nei’s nucleotide diversity (π)	0.026051	0.00558	0.02233	0.00593	0.33040	0.01056
Haplotype numeric (Hap)	43	24	36	19	59	54
Haplotype diversity (Hd)	0.91579	0.79500	0.92486	0.83520	0.97639	0.99562
Nucleotide diversity from S (θw)	0.09955	0.01603	0.03179	0.01676	0.04126	0.01794
Tajima’s *D*	−2.39947 *	−1.87578 ***	−0.92870 ****	−2.07658 ***	−0.63716 ****	−1.43591 ****
Fu and Li’s *D*	−1.53703 ****	−2.58454 ***	−0.33757 ****	−3.48112 **	−3.88613 **	−1.67899 ****
Fu’s *Fs*	−14.6930	−16.3734	−6.25117	−8.43211	−14.1927	−36.71492

^a^ 1: translation elongation factor 1-α gene (*EF1-α*); 2: calmodulin gene (*CAL*); 3: beta-tubulin 2 gene (*TUB2*); 4: histone-3 gene (*HIS*); 5: ribosomal internal transcribed spacer (ITS) region of ribosomal DNA (ITS1-5.8S-ITS2). **^b^** * *p* < 0.01; ** *p* < 0.02; *** *p* < 0.05; **** *p* > 0.10.

## Data Availability

Alignments generated during the current study are available in TreeBASE (accession http://purl.org/phylo/treebase/phylows/study/TB2:S26697 (accessed on 2 August 2020)). All sequence data are available in the NCBI GenBank, following the accession numbers in the manuscript.

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
