# Peer review of "Phylogenetic and Haplotype Network Analyses of Diaporthe eres Species in China Based on Sequences of Multiple Loci"

_biology, 2021, doi:10.3390/biology10030179_

Round 1
Reviewer 1 Report
The article provide insights on the eres complex. It is mandatory to say that all the study is based on sequencing done by different research group, so the data are not produced by the authors.
However the article deserves to be published only after considering a few minor inputs reported in the attached file.
regards.

Author Response
Reviewer 1
The article provide insights on the eres complex. It is mandatory to say that all the study is based on sequencing done by different research group, so the data are not produced by the authors.
Response: Thank you very much for your comment. In this study, for D. eres species boundaries, although some DNA sequences were obtained from corresponding database online, but we found the importance of phylogenetic analysis and pointed out the specific sequences are important or not important for the phylogenetic analysis. Such information could help scientists to enhance the analysis in future. Just based on these analyses, we found that three species that have been named as D. henanensis, D. lonicerae and D. rosicola should be the synonyms of D. eres. For haplotype network analyses, we used 47 sequences including new DNA sequence of MW221706 (from Citrus reticulata cv. Nanfengmiju in Jiangxi province, China), these sequences are produced by us (Table S1).
However the article deserves to be published only after considering a few minor inputs reported in the attached file.
- Why ITS has not been used for GCPSR analysis?
Response: Because we found that the ITS sequence was ineffective and unsuitable to speciate boundaries of Diaporthe species in phylogenetic informativeness (PI) analysis, so we excluded ITS for GCPSR analysis.
- As above, Why ITS has not been used for Haplotype analysis?
Response: Similar reason as we mentioned above, ITS has not been used in combined data with other sequences for haplotype analysis, however, we performed haplotype network analysis with each of locus (including ITS) which was shown in a supported figure S18, further indicating that ITS sequence is not suitable for analysis of Diaporthe diversity.
- For Diaporthe citri, please change the citation. Do not use Guarnaccia and Crous 2017.But use Guarnaccia and Crous 2018 –
Response: As we mentioned in the text, HIS=MF418281 of D. citri CBS 135422 was reported by Guarnaccia, V.; Crous, P.W. Emerging citrus diseases in Europe caused by species of Diaporthe. IMA Fungus 2017, 8, 317–334. And this sequence was not reported in Guarnaccia, V.; Crous, P.W. Species of Diaporthe on Camellia and Citrus in the Azores Islands. Phytopathol. Mediterr. 2018, 57, 307–319. So, we think the former one is more suitable.
Other changes:
Line 121–123: The sentence “Isolates were identified to be D. eres species based on multi-locus phylogenetic analyses with ITS, EF1-α, TUB2, CAL, and HIS in previous studies. Details of D. eres are shown in Supplementary Table S1.” was deleted according to your suggestion.
Reviewer 2 Report
The manuscript is OK. Should go through English editing though
Author Response
Reviewer 2
The manuscript is OK. Should go through English editing though
Response: Thank you very much for your positive comments, we really appreciate your encouragement. We re-checked the manuscript and revised some grammatical errors.
Reviewer 3 Report
The MS entitled “Phylogenetic and haplotype network analyses of Diaporthe eres species in China based on multiple sequences approach” by Chaisiri et al. represents a valuable contribution to the characterization of this species complex in China and contributes to the better general understanding of the boundaries of this species. The paper used a significant amount of data, and analyses are robust. Figures are nice and of good quality except Figure 2 in which the Bayesian posterior probabilities and bootstrap values for ML and MP cannot be read. Generally, the quality of the manuscript is greatly influenced by many errors in the English writing. It is strongly suggested to have the MS improved, to provide a better overall experience to readers.
The abstract and the introduction could be written better (see a few suggestions below). In the introduction section, the second paragraph (lines 59-72) should be completely re-written to describe in a more concise way the host range of Diaporthe eres.
Materials and methods section and results section are acceptable. However, the discussion section is rather a short review of very general topics, of other research groups findings rather than a concise comparison of the data from this study with previous research carried out using this plant pathogen, trying to assess relationships and the level of genetic diversity. Conclusions should be revised for clarity, for English language, and to acknowledge that the genes used in the current study were suggested to provide useful information by research carried out by other groups as well. Some suggestions regarding this section are provided below.
Selected, specific comments.
Please revise the title of the paper, it is written in a rather poor English. “Phylogenetic and Haplotype Network Analyses of Diaporthe eres Species in China Based on Multiple Sequences Approach”
Instead of “Based on Multiple Sequences Approach” I would suggest using something similar to “based on analyses of multiple loci”
Line 12
Replace “effects” with “affects”.
Lines 14-18
Phylogeny using the nucleotide or amino acid information of one or several genes is a molecular method. Please reformulate the sentences in lines 14-17 to reflect this fact. Also, mention that there are species of Diaporthe with unclear taxonomic status, some of which could be included in the D. eres complex of species. It is not clear from the simple summary why was needed to use molecular phylogeny.
Line 29.
Remove “In details”
Line 38
Likely that “previously research study” should actually be “previous research studies.”
Line 39
Please re-write the sentence : “ These isolates were obtained from different major plant species during 2006 to 2020.”
Do not use during, use “from … to”.
Lines 41-42.
Please re-write this sentence: “Haplotype networks were dispersal widespread within locality distribution.” This sentence is another example of improper use of English language. Disperse is different from dispersal.
.
Lines 51-52.
Please re-write this sentence without using the hyphens and make it more clear.
The genus Diaporthe (asexual morph, Phomopsis) represents a group of cosmopolitan species including plant-saprophytic, -endophytic, and -pathogenic ones with significant economic importance
Lines 354
valid for fungal identification from the genus at species level.
Reformulate this sentence to make it clear.
Line 336
Figure 5
Please change Locality distribution with Province distribution.
Line 367
Should be plural, criteria, not criterion.
Line 371
“Are” should be replaced by “were” or “have been”
Line 413.
Highly not high
Lines 422-423
“Current study provided the overview of D. eres on several plantation varieties and some useful information to identify this fungus.”
Why plantation varieties? Please reformulate this sentence.
Lines 425-427.
Please re-write this sentence and make it clear. Other research studies also indicated that EF1-α is a gene that is useful in phylogeny.
To identify D. eres species, we proposed using EF1-α locus for the best species delimitation which was supported by phylogenetic analyses and phylogenetic informativeness profiles.
Lines 436-438
Please revise the final sentence. In this phylogenetic study distribution and genetic variation was also studied so this sentence is not saying anything novel.
“However, D. eres in plants including studies of the distribution, genetic variation, and plant-D. eres interaction should be further conducted in the future.”
Author Response
Reviewer 3
The MS entitled “Phylogenetic and haplotype network analyses of Diaporthe eres species in China based on multiple sequences approach” by Chaisiri et al. represents a valuable contribution to the characterization of this species complex in China and contributes to the better general understanding of the boundaries of this species. The paper used a significant amount of data, and analyses are robust. Figures are nice and of good quality except Figure 2 in which the Bayesian posterior probabilities and bootstrap values for ML and MP cannot be read. Generally, the quality of the manuscript is greatly influenced by many errors in the English writing. It is strongly suggested to have the MS improved, to provide a better overall experience to readers.
Response: Thank you very much for your positive comments and good advice, we really appreciate your encouragement. For Figure 2, we revised it. Moreover, we have tried our best to revise the manuscript to improve readability as your suggestions, hope it can be acceptable.
The abstract and the introduction could be written better (see a few suggestions below). In the introduction section, the second paragraph (lines 59-72) should be completely re-written to describe in a more concise way the host range of Diaporthe eres.
Response: We really appreciate your advice and modified the second paragraph according to your suggestion as “Diaporthe eres was firstly collected with type specimen from Ulmus sp. in Germany. It was reported that D. eres could cause shoot blight on Acer pseudoplatanus [23] and Juglans cinerea [24]. It was also responsible for umbel browning and stem necrosis on Daucus carota [25], leaf necrosis on Hedera helix [26], fruit rot on Vitis sp. [27], stem canker and rootstock death on Malus spp. [28], etc. According to recent studies in China, it was responsible for branch canker, leaf blight, root rot etc on Cinnamomum camphora [29], Acanthopanax senticosus, Castanea mollissima, Melia azedarace, Rhododendron simsii, Sorbus sp. [20], Juglans regia [14,20], Polygonatum sibiricum [30], Photinia fraseri cv. Red Robin [31], Coptis chinensis [32], Acer palmatum [33], Pyrus sp. [16], Vitis sp. [19], Prunus persica [13] and Pinus albicaulis [34]. Among them, it often associates with many important economic trees, e.g., Camellia [35,36], Camptotheca [37], Citrus [18], grapevine [19,38], Japanese oak [39,40], kiwifruit [41], peach [13], pear [16], walnut [14], and so on”.
Materials and methods section and results section are acceptable. However, the discussion section is rather a short review of very general topics, of other research groups findings rather than a concise comparison of the data from this study with previous research carried out using this plant pathogen, trying to assess relationships and the level of genetic diversity. Conclusions should be revised for clarity, for English language, and to acknowledge that the genes used in the current study were suggested to provide useful information by research carried out by other groups as well. Some suggestions regarding this section are provided below.
Response: Thank you very much for your comments, we tried to revise it, hope it is better than before.
Selected, specific comments.
Please revise the title of the paper, it is written in a rather poor English. “Phylogenetic and Haplotype Network Analyses of Diaporthe eres Species in China Based on Multiple Sequences Approach”
Instead of “Based on Multiple Sequences Approach” I would suggest using something similar to “based on analyses of multiple loci”
Response: Thank you very much for your advice, we modified the title as “Phylogenetic and Haplotype Network Analyses of Diaporthe eres Species in China Based on Sequences of Multiple Loci” according to your suggestion. We did not use “analyses of multiple loci” but use “sequences of multiple loci” because the analyses have already been used in the sentence.
Line 12
Replace “effects” with “affects”.
Response: Did it according to your suggestion.
Lines 14-18
Phylogeny using the nucleotide or amino acid information of one or several genes is a molecular method. Please reformulate the sentences in lines 14-17 to reflect this fact. Also, mention that there are species of Diaporthe with unclear taxonomic status, some of which could be included in the D. eres complex of species. It is not clear from the simple summary why was needed to use molecular phylogeny.
Response: We appreciate you for the very important comment. The lines 14-18 were revised as “In general, morphological and molecular characterization using multiple loci sequences were performed for the identification of Diaporthe species. However, there are morphological differences due to culture conditions, and the taxonomy of species of Diaporthe is unclear because multi-locus phylogenetic analysis could obtain different results with different loci sequences” according to your suggestions.
Line 29.
Remove “In details”
Response: Did it according to your suggestion.
Line 38
Likely that “previously research study” should actually be “previous research studies.”
Response: We agree with you and modified it as “previous studies”.
Line 39
Please re-write the sentence : “These isolates were obtained from different major plant species during 2006 to 2020.”
Do not use during, use “from … to”.
Response: The sentence has been revised as “These isolates were obtained from different major plant species from 2006 to 2020.”
Lines 41-42.
Please re-write this sentence: “Haplotype networks were dispersal widespread within locality distribution.” This sentence is another example of improper use of English language. Disperse is different from dispersal.
Response: The whole sentence has been revised as “Haplotype networks were widely dispersed and not uniquely correlated to specific populations”.
Lines 51-52.
Please re-write this sentence without using the hyphens and make it more clear.
The genus Diaporthe (asexual morph, Phomopsis) represents a group of cosmopolitan species including plant-saprophytic, -endophytic, and -pathogenic ones with significant economic importance
Response: The sentence has been revised as “ The genus Diaporthe (asexual morph, Phomopsis) represents a group of cosmopolitan species including saprophytic, endophytic, and pathogenic ones on different plants”.
Lines 354
valid for fungal identification from the genus at species level.
Reformulate this sentence to make it clear.
Response: The sentence was revised as “ Next generation sequencing (NGS) technology as DNA barcoding is highly efficient, more accurate, thus valid for fungal identification at species-level”.
Line 336
Figure 5
Please change Locality distribution with Province distribution.
Response: Did it according to your suggestion.
Line 367
Should be plural, criteria, not criterion.
Response: The sentence has been revised as “Many Diaporthe spp. were classified based on different criteria” according to your suggestion.
Line 371
“Are” should be replaced by “were” or “have been”
Response: We changed “are” to “were” according to your suggestion.
Line 413.
Highly not high
Response: Did it as you suggested.
Lines 422-423
“Current study provided the overview of D. eres on several plantation varieties and some useful information to identify this fungus.”
Why plantation varieties? Please reformulate this sentence.
Response: The sentence has been revised as “Current study provided an overview of D. eres on several plant varieties and some valuable knowledge to identify this fungus”.
Lines 425-427.
Please re-write this sentence and make it clear. Other research studies also indicated that EF1-α is a gene that is useful in phylogeny.
To identify D. eres species, we proposed using EF1-α locus for the best species delimitation which was supported by phylogenetic analyses and phylogenetic informativeness profiles.
Response: The sentence has been revised as “To identify D. eres species, we proposed using EF1-α locus for the best species delimitation which was supported by phylogenetic analyses and phylogenetic informativeness profiles” according to your suggestion.
Lines 436-438
Please revise the final sentence. In this phylogenetic study distribution and genetic variation was also studied so this sentence is not saying anything novel.
“However, D. eres in plants including studies of the distribution, genetic variation, and plant-D. eres interaction should be further conducted in the future.”
Response: We modified it as “In the future, studies such as the evolution of D. eres, plant-D. eres interaction should be further conducted” according to your suggestion.
Round 2
Reviewer 3 Report
The authors marginally improved the manuscript. Several corrections still have to be made throughout the MS to improve the English language and readability/clarity of the text.
Below are listed just a few, more important suggestions:
Line 15-17. Please re-write the sentence as its message is not clear.
… and the taxonomy of species of Diaporthe is unclear because multi-locus phylogenetic analysis could obtain different results with different loci sequences.
What do you want to say? That phylogeny based on different genes gave different tree topologies or they failed to indicate the real relationship which can be revealed only by concatenating several genes (loci)?
Lines 365-366
Replace “applied by us are not” with “used in the current study by us were”
Also, add “the” and remove “being”.
The sentence should read something like:
“Thus, the species used in the current by us were not always the same as those recovered by the single locus MegaBlast…”
Line 416-417.
The revision of the sentence is not acceptable because you again failed to indicate that other researchers also found that EF1 is important in discriminating D. eres species. Also, the sentence reads quite badly in English.
I suggest writing a sentence something like:
“Phylogenetic analyses and phylogenetic informativeness profiles reported in this study revealed that for D. eres species identification and delimitation the usage of the EF1-α locus represents the optimal alternative, proposition that is also supported by previous research [43,47,91,98].”
Lines 418-419
Please revise the sentence: “Moreover, it was revealed that the ITS region impeded reliable species recognition for D. eres species complex”
Maybe you could write something like:
“Moreover, our analyses revealed that the usage of the ITS region hampers proper species recognition within the D. eres species complex”
Line 422
Replace “supports” with “support”.
.. support from population genetic parameters and neutrality of statistic informative values
Author Response
The authors marginally improved the manuscript. Several corrections still have to be made throughout the MS to improve the English language and readability/clarity of the text.
Response: Thank you very much for your encouragement, We made another round of corrections mainly based on your suggestions.
Below are listed just a few, more important suggestions:
Line 15-17. Please re-write the sentence as its message is not clear.
… and the taxonomy of species of Diaporthe is unclear because multi-locus phylogenetic analysis could obtain different results with different loci sequences.
What do you want to say? That phylogeny based on different genes gave different tree topologies or they failed to indicate the real relationship which can be revealed only by concatenating several genes (loci)?
Response: Thank you very much, we rewrote it as “and the taxonomy of species of Diaporthe is unclear because the phylogeny based on different genes gave different tree topologies” according to your suggestion.
Lines 365-366
Replace “applied by us are not” with “used in the current study by us were”
Also, add “the” and remove “being”.
The sentence should read something like:
“Thus, the species used in the current by us were not always the same as those recovered by the single locus MegaBlast…”
Response: Thank you very much for your nice and clear advice to revise this sentence. We modified this sentence as “Thus, the species used in the current study by us were not always the same as those recovered by the single locus MegaBlast search in NCBI” according to your suggestion.
Line 416-417.
The revision of the sentence is not acceptable because you again failed to indicate that other researchers also found that EF1 is important in discriminating D. eres species. Also, the sentence reads quite badly in English.
I suggest writing a sentence something like:
“Phylogenetic analyses and phylogenetic informativeness profiles reported in this study revealed that for D. eres species identification and delimitation the usage of the EF1-α locus represents the optimal alternative, proposition that is also supported by previous research [43,47,91,98].”
Response: We agree, and rewrote it as “Phylogenetic analyses and phylogenetic informativeness profiles reported in this study revealed that for D. eres species identification and delimitation, the usage of the EF1-α locus represents the optimal alternative, proposition that is also supported by previous studies [43,47,91,98]” according to your suggestion.
Lines 418-419
Please revise the sentence: “Moreover, it was revealed that the ITS region impeded reliable species recognition for D. eres species complex”
Maybe you could write something like:
“Moreover, our analyses revealed that the usage of the ITS region hampers proper species recognition within the D. eres species complex”
Response: Thank you very much, we modified the sentence as “Moreover, our analyses revealed that the usage of the ITS region hampers proper species recognition within the D. eres species complex” accordingly.
Line 422
Replace “supports” with “support”.
.. support from population genetic parameters and neutrality of statistic informative values
Response: We did it according to your suggestion.
Again we really appreciate you for providing these valuable suggestions.
This manuscript is a resubmission of an earlier submission. The following is a list of the peer review reports and author responses from that submission.
Round 1
Reviewer 1 Report
I renovate the same comments previously included for your submission in another journal of the same group MDPI.
I'm really disappointed to see that previous comments were nor considered.
The article put together several reference sequences deposited in the past from especially other studies. Overall the results are well presented, clear and well analysed.
-The authors should write a more complete and detailed introduction which is quite vague at the moment. The authors could consider very recent papers about Diaporthe taxonomy and diseases such as:
Lesuthu, P., Mostert, L., Spies, C. F., Moyo, P., Regnier, T., & Halleen, F. (2019). Diaporthe nebulae sp. nov. and first report of D. cynaroidis, D. novem, and D. serafiniae on Grapevines in South Africa. Plant disease, 103(5), 808-817.
Guarnaccia, V., & Crous, P. W. (2018). Species of Diaporthe on Camellia and Citrus in the Azores Islands.
Zapata, M., Palma, M. A., Aninat, M. J., & Piontelli, E. (2020). Polyphasic studies of new species of Diaporthe from native forest in Chile, with descriptions of Diaporthe araucanorum sp. nov., Diaporthe foikelawen sp. nov. and Diaporthe patagonica sp. nov. International Journal of Systematic and Evolutionary Microbiology, 70(5), 3379-3390.
Moreover, the discussion and conclusions are redundant and repetitive, the authors should fix this aspect.
Reviewer 2 Report
The manuscript describes the resolution of the phylogeny of Diaporthe eres. Although the methods used were solid it is a very limited investigation. I do not see here enough interesting material and answer to important biological questions in fungal biology. Therefore I do not think the manuscript should be accepted.